# Rare Earth Element Geochemistry of Late Cenozoic Island Carbonates in the South China Sea

Xiao-Feng Liu [1], Shikui Zhai [1,*], Xi-Kai Wang [2], Xinyu Liu [3] and Xiao-Ming Liu [2,*]

1   College of Marine Geosciences, Ocean University of China, Qingdao 266100, China; xfliu2012@163.com
2   Department of Earth, Marine and Environmental Sciences, University of North Carolina, Chapel Hill, NC 27599, USA; xikai@live.unc.edu
3   Zhanjiang Branch of China National Offshore Oil Corporation (CNOOC) Limited, Zhanjiang 524057, China; liuxy5@cnooc.com.cn
*   Correspondence: zhai2000@ouc.edu.cn (S.Z.); xiaomliu@unc.edu (X.-M.L.)

**Abstract:** Marine carbonates, precipitating from seawater through inorganic geochemical and biogeochemical processes, are considered to have recorded the seawater geochemical signatures reflecting the marine environmental conditions during their formation. However, they are susceptible to post-depositional diagenetic alteration. The redox conditions and chemical composition of the diagenetic fluid may be different from those of the overlying seawater. Therefore, assessing whether carbonate rocks that have experienced variable diagenesis could still preserve primary seawater geochemistry is a prerequisite before inferring ancient marine environments using geochemical tracers such as the cerium anomaly (Ce/Ce*). Here, we investigate rare earth elements plus yttrium (REY) geochemical features of reefal carbonates from the XK-1 core in the Xisha Islands of the South China Sea. We aim to evaluate whether island carbonates have the potential to preserve reliable primary seawater REY geochemical characteristics after experiencing meteoric diagenesis, marine burial diagenesis, or dolomitization. The results show that even after variable diagenetic alteration, all carbonate samples exhibit seawater-like REY patterns, which are characterized by negative Ce anomalies (Ce/Ce* < 1), distinctively high Y/Ho ratios (>44), and uniform depletion of light rare earth elements (LREE) relative to heavy rare earth elements (HREE) ((Pr/Yb)$_N$ < 1). This suggests that the original seawater REY signatures are retained, regardless of varying degrees of changes in the mineralogical composition, diagenetic fluid composition, and redox state. The unmodifiable REY characteristics in carbonates during diagenesis can be attributed to three aspects: (1) during meteoric diagenesis, the low REY content of meteoric fluids and the short-term reactions between fluids and carbonates make it difficult to significantly alter the REY composition of carbonates; (2) during marine burial diagenesis, the ubiquitous cementation creates a relatively closed environment that facilitates the inheritance of REY signatures from primary carbonates; (3) during dolomitization, the dolomitizing fluids derived from penecontemporaneous seawater would not destroy but rather promote the preservation of the original seawater REY signatures in dolostones. The Ce/Ce* variations indicate that the Xisha carbonates have been deposited under constantly oxic conditions since the Neogene, consistent with paleontological and redox-sensitive element geochemical evidence.

**Keywords:** rare earth elements; cerium anomaly; marine carbonates; carbonate diagenesis; Xisha Islands

## 1. Introduction

Marine carbonate minerals, such as aragonite and calcite, due to precipitating from seawater through inorganic geochemical or biogeochemical processes, can record paleo-seawater geochemical information during their formation. Thus, marine carbonates have been widely used to indicate marine environmental changes in geologic history [1–4]. REY includes rare earth elements (REEs) and yttrium (Y), where REEs refer to the lanthanides with the atomic number from 57 ($^{57}$La) to 71 ($^{71}$Lu). Owing to their similar atomic structures

and ion radiuses, all elements in REY have similar geochemical properties and usually coexist in nature. They behave conservatively and do not easily migrate. Hence, the REY signatures in carbonates are widely used to trace the geochemical properties of the surrounding fluids (such as seawater and pore water) and environmental conditions during primary carbonate deposition [1,2,4–7].

Diagenesis, however, could result in the dissolution and recrystallization of primary minerals in the post-depositional process, modifying the original seawater REY signatures preserved in carbonates [3,8,9]. It is still controversial whether the transformation from calcite to dolomite during dolomitization leads to a significant loss of primary REY signatures [2,3,6,10,11]. In addition, the mixing of terrigenous materials into marine carbonates will also reduce the reliability of using REY proxies to trace the primary marine environment [12,13]. Previous studies [14–19] have shown that the Cenozoic island carbonates have the following advantages: (1) the genesis is relatively simple as they are uniformly formed in shallow marine environments; (2) they are relatively young and are unaffected by multiple geological processes over a long geologic history; (3) they are less contaminated by terrigenous input, being far away from the continent; (4) the diagenetic environments and dolomitization types are easy to be identified. Therefore, the Cenozoic island carbonates are ideal materials for evaluating the effects of early diagenesis, including dolomitization, on the original seawater REY signatures preserved in carbonates.

The reefal carbonates in the Xisha Islands of the South China Sea (abbreviated as the Xisha carbonates) have been deposited since the Neogene, and contain several highly pure calcite and dolomite intervals [18,20–24]. Here, we analyzed the mineralogical composition and elemental concentration of carbonate samples from the XK-1 core in the Xisha Islands. Combining the petrographic features, mineralogical composition, and published carbon and oxygen isotopic data, we identified diagenesis types for the Xisha carbonates. Then, we evaluated the possible effects of diagenetic alteration on the Xisha carbonate samples in the post-depositional process. Finally, by comparing the REY characteristics between carbonate samples and modern shallow seawater in the South China Sea, we evaluated whether the samples that had experienced variable diagenesis could still preserve the original seawater REY signals. We also showed the potential of the Ce anomaly (Ce/Ce*) in carbonate rocks as a tracer for paleo-redox conditions.

## 2. Geological Setting

The Xisha Islands (15°43′–17°07′ N, 111°11′–112°54′ E) are located at the continental margin of the northwestern South China Sea. The archipelago consists of more than 40 reefs, with a sea area of approximately $1.5 \times 10^4$ km$^2$ and a land area of approximately 8 km$^2$ [25] (Figure 1a). Based on the modern distribution characteristics, the reefs in the Xisha Islands can be classified into large atolls (Xuande atoll, Yongle atoll, Dongdao atoll, and Huaguang atoll), medium atolls (Beijiao atoll, Yuzhuo atoll, Langhua atoll, and Panshiyu atoll), and small reefs (Jinyintai reef and Zhongjiantai reef). The Xisha carbonate platform is located on the basement of the Xisha uplift, which is composed of the Precambrian granite gneiss and Mesozoic granite, with carbonate strata approximately 1260 m thick [25–28] (Figure 1b). During the transitional period from the Oligocene to the Miocene, due to the influence of seafloor spreading and regional crustal extension in the South China Sea [29,30], the Xisha basement began to subside, transgression occurred, and reefs developed on the basement. Subsequently, under the post-rifting thermal subsidence prevalent in the northern continental margin of the South China Sea, the Xisha basement continued to subside [21,31,32]. At the same time, with the fluctuation of relative sea level, reefs grew or drowned, gradually forming a carbonate platform surrounded by basins or depressions with a unified water depth of more than 1000 m [21]. According to detailed stratigraphic studies of six scientific drilling cores (XY-1, XC-1, XY-2, XS-1, XK-1, and CK-2) in the Xisha Islands, the carbonate strata can be divided into the Sanya formation in the early Miocene, Meishan formation in the middle Miocene, Huangliu formation in the late Miocene, Yinggehai formation in the Pliocene, and Ledong formation in the

Quaternary [20,21]. The XK-1 well is located on the Xuande Atoll (Figure 1a). It is 1368.02 m deep, penetrating the basement of the Xisha carbonate platform (Figure 1b). The carbonate succession is 1257.52 m thick and the recovery rate of the XK-1 core is up to 80%. The most pronounced feature of the carbonate succession is the interbedded distribution of thick-bedded limestone and dolostone (Figure 1b). In addition, the core is rich in paleontological fossils, such as red algae, corals, foraminifera, echinoderms, brachiopods, gastropods, bivalves, green algae, and calcareous nannofossils [33–36].

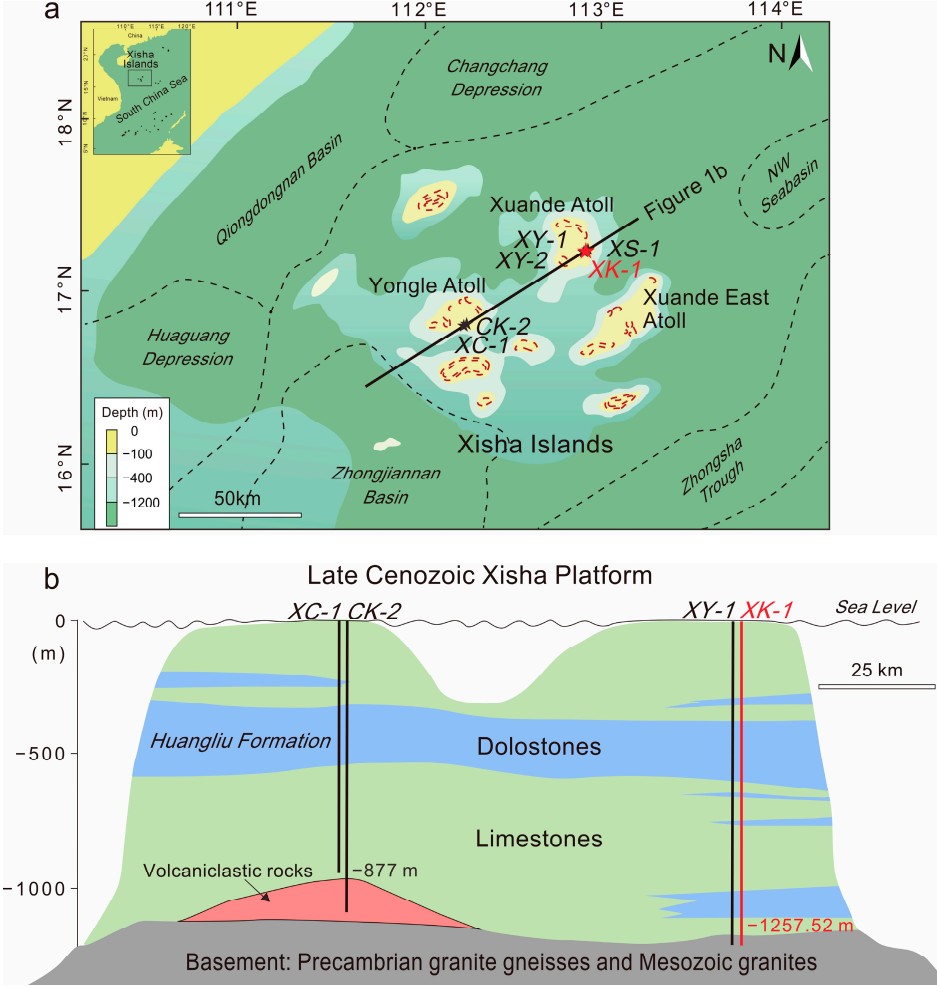

**Figure 1.** (**a**) Location of the Xisha Islands in the South China Sea, and distribution of wells. (**b**) Cross-section through the Xisha carbonate platform. See Figure 1a for the location. Figures are modified from [18,21].

## 3. Materials and Methods

Fifty-two carbonate samples from the XK-1 core were selected for mineralogical and element geochemical analyses, based on the equidistant sampling principle.

### 3.1. X-ray Diffraction Analysis

Mineral phase identification was conducted in the Open Laboratory, Qingdao Institute of Bioenergy and Bioprocess Technology, Chinese Academy of Sciences. First, the sample was ground to a particle size of less than 74 μm using a clean agate mortar. After drying at 100 °C, the sample powder was then carefully scraped into the cavity of an aluminum sample holder and gently pressed with a glass slide to counteract the tendency of the particles to be parallel to the glass surface. This produces randomly oriented aggregates and allows for the quantitative peak fitting. Finally, X-ray diffraction (XRD) measurement

was performed on a Bruker D8 Advance X-ray diffractometer, using a rotating Ni-filtered Cu anode target X-ray source ($\lambda = 0.15406$ nm). Test conditions were as follows: voltage of 40 kV, current of 40 mA, scanning increment of $0.02°$ ($2\theta$), scanning rate of $4°/\text{min}$ ($2\theta$), scanning range of 15–80° ($2\theta$). Before performing the test, the instrument was calibrated using standard materials for the accuracy of the peak determination. MDI Jade 6.5 software was used for data processing. The relative percentages of carbonate minerals (aragonite, calcite and dolomite) were calculated based on the (104) peak fitting results (aragonite: $2\theta = 25.50°$–26.50°, d = 3.390–3.400 Å; calcite: $2\theta = 29.25°$–29.80°, d = 2.995–3.035 Å; dolomite: $2\theta = 30.58°$–31.28°, d = 2.854–2.912 Å) [37].

### 3.2. Elemental Analysis

Elemental analysis was carried out in the Plasma Mass Spectrometry Laboratory of the University of North Carolina at Chapel Hill (UNC-CH), Chapel Hill, NC, USA. Approximately 5–10 mg of dried powder sample was weighed and placed into a 50 mL centrifuge tube. Before dissolving, all samples were subject to a pre-leaching step following a method modified from [38]: the sample was mixed with 1 M ammonium acetate (pH = 7) for 1 h and rinsed with Milli-Q water (18.2 M$\Omega$) three times to remove the exchangeable ions. The sample was then dissolved with 30 mL of 0.05 M HCl for 12 h. At each step, powder and solution were well mixed using a vortex mixer and a reciprocal shaker to improve reaction efficiency. The above method can achieve complete carbonate dissolution while avoiding contamination from non-carbonate components [39–41]. After dissolution, the sample was centrifuged at 5000 rpm for 10 min. The supernatant fluid was filtered into a Teflon beaker using a metal-free filter with pore sizes of 0.22 μm before it was dried down and re-dissolved by double-distilled concentrated $HNO_3$ several times. Finally, the carbonate leachate was evaporated and loaded into 2% (*v/v*) $HNO_3$ for subsequent elemental concentration determinations.

The elemental concentrations were measured on an Agilent[TM] 7900 quadrupole inductively coupled plasma mass spectrometer (Q-ICP-MS, Agilent[TM]). A series of multi-element reference solutions of known concentrations (prepared by the ICP standard solutions, Inorganic Ventures[TM]) were used as external standards to acquire the concentration calibration curves. Internal standard solution, including 100 ppb Be, Ge, Ph, In, Ir, and Bi, was used to correct instrument drift. Data quality was monitored through repeated analyses of the international certified reference material NIST SRM 1d (limestone) obtained from the United States Geological Survey. The analytical accuracies for REEs and other elements (Al, Mn, Fe, Sr, Y, and Th) were mostly better than 10%.

### 3.3. Expression of REY Proxies

The total REY concentration is expressed as $\Sigma$REY. When plotting REY distribution patterns (Y is usually inserted between Dy and Ho), the measured REY contents of carbonates are commonly normalized to Post-Archean Australian Shale (PAAS) [42]. Ce/Ce* (Ce anomaly), $(\text{Pr/Yb})_N$ (PAAS-normalized Pr/Yb ratio), and Y/Ho are typical REY proxies in carbonates and seawater [7]. The Ce/Ce* value is calculated based on the following equation [43]:

$$\text{Ce/Ce*} = \text{Ce}_N / (\text{Pr}_N{}^2 / \text{Nd}_N)$$

where the subscript N represents the PAAS-normalized value. When the calculated Ce/Ce* value is between 0.9 and 1.1, it is considered non-anomalous. In addition, $(\text{Pr/Yb})_N$ is the PAAS-normalized Pr/Yb ratio. Y/Ho is the content ratio of Y to Ho in the sample.

### 3.4. Diagenesis Identification

The petrographic, mineralogical, and carbon and oxygen isotopic signatures were combined to identify the diagenetic features experienced by the XK-1 core carbonates. Petrographic photographs, including photographs of hand specimens, optical micrographs, scanning electron micrographs (SEM photographs), and cathodoluminescence (CL) images, were provided by the Zhanjiang Branch of China National Offshore Oil Corporation

(CNOOC) Limited. The mineralogical data were integrated from this study and our previous studies [22,23]. Carbon and oxygen isotopic data were compiled from previous research [44–46].

## 4. Results

All measured data, including mineralogical composition and elemental concentrations of carbonate samples from the XK-1 core, are shown in Table 1.

### 4.1. Mineralogical Composition

The core is mainly composed of carbonates that contain aragonite (ARA), high-magnesium calcite (HMC), low-magnesium calcite (LMC), and dolomite (DOL) (Figure 2). Among them, DOL and LMC are predominant, and they appear alternately. HMC is distributed intensively above the depth of 22 m. ARA is primarily distributed at depths of 0–36 m and 207–230 m. Terrigenous minerals such as feldspar, quartz, kaolinite, smectite, and mica are found mostly between depths of 1216–1257.52 m.

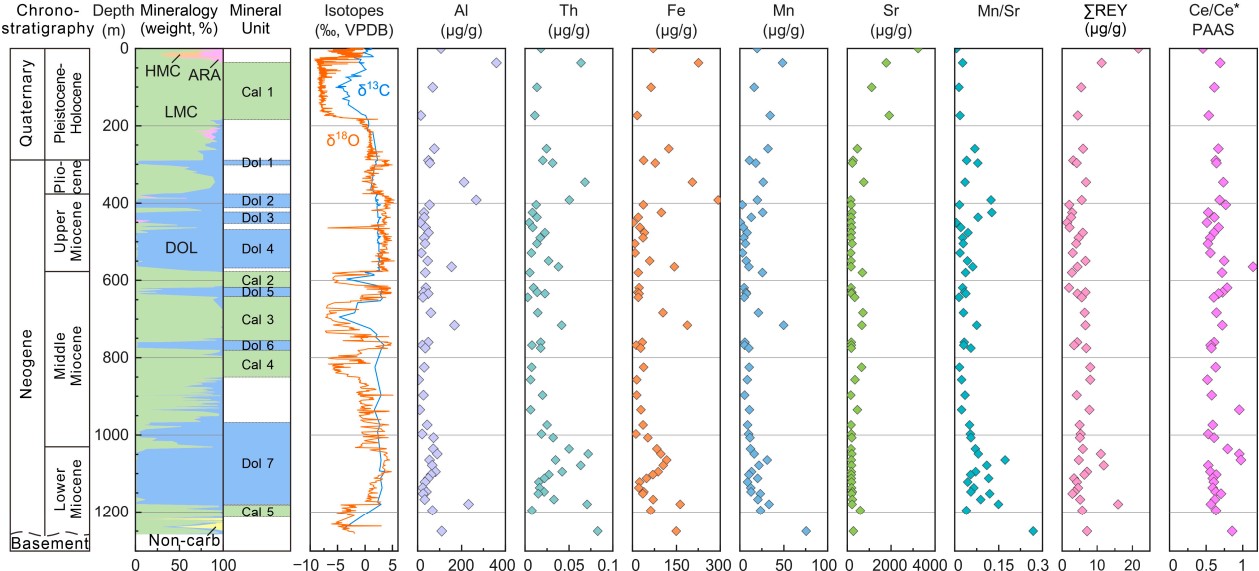

**Figure 2.** The stratigraphic division, mineralogical composition, and geochemical composition of the XK-1 core carbonates. The stratigraphic division scheme is from [47], mineralogical data are integrated from this study and our previous studies [22,23], and $\delta^{13}$C and $\delta^{18}$O data on the Vienna Pee Dee Belemnite (VPDB) scale are from [44–46]. ARA: aragonite, HMC: high-magnesium calcite, LMC: low-magnesium calcite, DOL: dolomite, and Non-carb: non-carbonate minerals. Cal 1–5 denote five calcite units with calcite content higher than 80%, and Dol 1–7 denote seven dolomite units with dolomite content higher than 80%. Ce/Ce* values are normalized to Post-Archean Australian Shale (PAAS) [42].

In this paper, samples with calcite or dolomite content of more than 90% are defined as calcite samples or dolomite samples, respectively. Except for one surficial mixture sample (depth of 0.3 m), the other 51 samples are composed mainly of pure calcite or dolomite. The specific distribution locations of the samples in the core column are shown in Figure 2. Combining the mineralogical data reported in this study with our previously published data [22,23], five calcite units and seven dolomite units are delineated (Figure 2). The division standard of the mineral units in the XK-1 core is that the relative content of one mineral is successively higher than 80% in a certain depth range. Calcite unit 1: depths of 35.4–180.3 m; calcite unit 2: 564.96–615.2 m; calcite unit 3: 636.96–758.4 m; calcite unit 4: 779.8–849.8 m; calcite unit 5: 1184–1211 m. Dolomite unit 1: 289.3–312.3 m; dolomite unit 2: 373.3–412.7 m; dolomite unit 3: 423.7–450.6 m; dolomite unit 4: 469.7–564.96 m; dolomite unit 5: 615.2–636.96 m; dolomite unit 6: 758.4–779.8 m; dolomite unit 7: 966.8–1179.5 m.

**Table 1.** Mineralogical and elemental data of the XK-1 core carbonates.

| Depth m | LMC % | DOL % | Mineral Unit | Al µg/g | Mn µg/g | Fe µg/g | Sr µg/g | Y µg/g | La µg/g | Ce µg/g | Pr µg/g | Nd µg/g | Sm µg/g | Eu µg/g | Gd µg/g | Tb µg/g | Dy µg/g | Ho µg/g | Er µg/g | Tm µg/g | Yb µg/g | Lu µg/g | Th µg/g | ∑REY µg/g | Ce/Ce* | $(Pr/Yb)_N$ | Y/Ho | Mn/Sr |
|---|---|---|---|---|---|---|---|---|---|---|---|---|---|---|---|---|---|---|---|---|---|---|---|---|---|---|---|---|
| 0.3 | HMC: 43 LMC: 31 ARA: 26 | | — | 106 | 20 | 71 | 3231 | 11.23 | 2.36 | 1.47 | 0.49 | 2.36 | 0.59 | 0.14 | 0.83 | 0.12 | 0.85 | 0.17 | 0.53 | 0.06 | 0.49 | 0.05 | 0.02 | 21.77 | 0.46 | 0.31 | 65 | 0.01 |
| 37.0 | 100 | 0 | Cal 1 | 359 | 49 | 225 | 1777 | 5.07 | 1.36 | 1.42 | 0.29 | 1.34 | 0.31 | 0.08 | 0.40 | 0.06 | 0.38 | 0.08 | 0.23 | 0.03 | 0.23 | 0.02 | 0.06 | 11.28 | 0.69 | 0.40 | 67 | 0.03 |
| 100.3 | 100 | 0 | Cal 1 | 69 | 16 | 64 | 1113 | 3.08 | 0.48 | 0.39 | 0.10 | 0.48 | 0.12 | 0.03 | 0.18 | 0.03 | 0.20 | 0.04 | 0.14 | 0.02 | 0.14 | 0.02 | 0.01 | 5.44 | 0.61 | 0.22 | 74 | 0.01 |
| 173.3 | 100 | 0 | Cal 1 | 15 | 34 | 16 | 1913 | 2.51 | 0.43 | 0.28 | 0.08 | 0.39 | 0.10 | 0.03 | 0.15 | 0.02 | 0.16 | 0.03 | 0.11 | 0.01 | 0.11 | 0.01 | 0.01 | 4.44 | 0.54 | 0.22 | 73 | 0.02 |
| 259.3 | 90.4 | 9.6 | — | 76 | 32 | 124 | 464 | 2.82 | 0.65 | 0.63 | 0.14 | 0.69 | 0.17 | 0.04 | 0.23 | 0.03 | 0.23 | 0.05 | 0.14 | 0.02 | 0.15 | 0.01 | 0.02 | 6.00 | 0.67 | 0.31 | 61 | 0.07 |
| 289.3 | 4.4 | 95.6 | Dol 1 | 49 | 11 | 38 | 267 | 1.46 | 0.41 | 0.32 | 0.08 | 0.37 | 0.09 | 0.02 | 0.12 | 0.02 | 0.12 | 0.02 | 0.07 | 0.01 | 0.07 | 0.01 | 0.02 | 3.18 | 0.63 | 0.35 | 62 | 0.04 |
| 296.3 | 0 | 100 | Dol 1 | 55 | 18 | 78 | 229 | 1.63 | 0.58 | 0.47 | 0.11 | 0.55 | 0.13 | 0.03 | 0.17 | 0.03 | 0.18 | 0.03 | 0.10 | 0.01 | 0.09 | 0.01 | 0.03 | 4.14 | 0.64 | 0.38 | 48 | 0.08 |
| 346.3 | 90.2 | 9.8 | — | 212 | 27 | 205 | 748 | 2.70 | 0.91 | 1.01 | 0.20 | 0.91 | 0.21 | 0.05 | 0.25 | 0.03 | 0.22 | 0.04 | 0.13 | 0.01 | 0.12 | 0.01 | 0.07 | 6.80 | 0.74 | 0.50 | 65 | 0.04 |
| 392.1 | 2.2 | 97.8 | Dol 2 | 267 | 20 | 293 | 162 | 2.02 | 0.83 | 0.79 | 0.17 | 0.79 | 0.19 | 0.05 | 0.24 | 0.04 | 0.23 | 0.04 | 0.13 | 0.01 | 0.11 | 0.01 | 0.05 | 5.63 | 0.69 | 0.48 | 47 | 0.12 |
| 404.7 | 0 | 100 | Dol 2 | 54 | 3 | 37 | 171 | 0.71 | 0.31 | 0.31 | 0.06 | 0.28 | 0.07 | 0.02 | 0.08 | 0.01 | 0.08 | 0.02 | 0.04 | 0.005 | 0.04 | 0.004 | 0.01 | 2.02 | 0.77 | 0.50 | 46 | 0.02 |
| 424.7 | 0 | 100 | Dol 3 | 30 | 26 | 99 | 205 | 1.34 | 0.43 | 0.24 | 0.07 | 0.33 | 0.08 | 0.02 | 0.10 | 0.01 | 0.10 | 0.02 | 0.06 | 0.01 | 0.06 | 0.01 | 0.01 | 2.88 | 0.53 | 0.36 | 70 | 0.13 |
| 436.6 | 0 | 100 | Dol 3 | 30 | 13 | 20 | 165 | 0.96 | 0.42 | 0.32 | 0.08 | 0.37 | 0.08 | 0.02 | 0.11 | 0.02 | 0.10 | 0.02 | 0.06 | 0.01 | 0.05 | 0.01 | 0.01 | 2.60 | 0.61 | 0.50 | 52 | 0.08 |
| 450.6 | 0 | 100 | Dol 3 | 12 | 1 | 3 | 170 | 0.62 | 0.22 | 0.14 | 0.04 | 0.19 | 0.04 | 0.01 | 0.06 | 0.01 | 0.05 | 0.01 | 0.03 | 0.004 | 0.03 | 0.003 | 0.005 | 1.47 | 0.51 | 0.41 | 59 | 0.004 |
| 462.7 | 2 | 98 | Dol 4 | 36 | 4 | 26 | 188 | 0.92 | 0.29 | 0.23 | 0.05 | 0.25 | 0.06 | 0.01 | 0.08 | 0.01 | 0.07 | 0.01 | 0.04 | 0.01 | 0.04 | 0.004 | 0.01 | 2.08 | 0.67 | 0.40 | 64 | 0.02 |
| 476.7 | 0 | 100 | Dol 4 | 49 | 8 | 41 | 181 | 2.66 | 0.82 | 0.61 | 0.15 | 0.71 | 0.17 | 0.04 | 0.22 | 0.03 | 0.21 | 0.04 | 0.12 | 0.01 | 0.10 | 0.01 | 0.02 | 5.89 | 0.61 | 0.45 | 66 | 0.04 |
| 489.9 | 0 | 100 | Dol 4 | 30 | 5 | 36 | 170 | 2.07 | 0.69 | 0.44 | 0.12 | 0.58 | 0.13 | 0.03 | 0.18 | 0.03 | 0.17 | 0.03 | 0.10 | 0.01 | 0.08 | 0.01 | 0.02 | 4.67 | 0.55 | 0.46 | 61 | 0.03 |
| 504.8 | 0 | 100 | Dol 4 | 34 | 6 | 8 | 212 | 1.43 | 0.66 | 0.47 | 0.13 | 0.64 | 0.15 | 0.03 | 0.17 | 0.02 | 0.15 | 0.03 | 0.07 | 0.01 | 0.06 | 0.01 | 0.01 | 4.03 | 0.53 | 0.72 | 56 | 0.03 |
| 528.8 | 0 | 100 | Dol 4 | 16 | 3 | 9 | 162 | 1.23 | 0.47 | 0.30 | 0.08 | 0.41 | 0.10 | 0.02 | 0.13 | 0.02 | 0.11 | 0.02 | 0.06 | 0.01 | 0.05 | 0.01 | 0.01 | 3.02 | 0.56 | 0.55 | 57 | 0.02 |
| 549.8 | 0 | 100 | Dol 4 | 46 | 7 | 59 | 167 | 3.30 | 0.89 | 0.66 | 0.13 | 0.66 | 0.15 | 0.04 | 0.24 | 0.03 | 0.24 | 0.05 | 0.14 | 0.02 | 0.12 | 0.01 | 0.03 | 6.68 | 0.75 | 0.35 | 70 | 0.04 |
| 565.0 | 1.9 | 98.1 | Dol 4 | 155 | 10 | 143 | 169 | 1.32 | 0.63 | 1.00 | 0.13 | 0.62 | 0.14 | 0.03 | 0.16 | 0.02 | 0.14 | 0.02 | 0.07 | 0.01 | 0.06 | 0.01 | 0.04 | 4.36 | 1.14 | 0.69 | 53 | 0.06 |
| 579.8 | 100 | 0 | Cal 2 | 35 | 26 | 20 | 689 | 1.42 | 0.32 | 0.28 | 0.06 | 0.30 | 0.07 | 0.02 | 0.10 | 0.01 | 0.09 | 0.02 | 0.05 | 0.01 | 0.05 | 0.005 | 0.01 | 2.79 | 0.72 | 0.37 | 82 | 0.04 |
| 618.8 | 2.1 | 97.9 | Dol 5 | 37 | 5 | 23 | 172 | 0.86 | 0.26 | 0.24 | 0.05 | 0.24 | 0.06 | 0.01 | 0.07 | 0.01 | 0.07 | 0.01 | 0.04 | 0.004 | 0.03 | 0.003 | 0.01 | 1.96 | 0.79 | 0.52 | 68 | 0.03 |
| 630.4 | 2 | 98 | Dol 5 | 28 | 7 | 18 | 207 | 2.61 | 1.08 | 0.86 | 0.18 | 0.91 | 0.21 | 0.05 | 0.26 | 0.04 | 0.21 | 0.04 | 0.10 | 0.01 | 0.08 | 0.01 | 0.01 | 6.65 | 0.72 | 0.71 | 67 | 0.03 |
| 633.9 | 2.1 | 97.9 | Dol 5 | 48 | 8 | 22 | 197 | 1.73 | 0.66 | 0.56 | 0.13 | 0.61 | 0.15 | 0.04 | 0.17 | 0.02 | 0.15 | 0.03 | 0.07 | 0.01 | 0.06 | 0.01 | 0.02 | 4.38 | 0.67 | 0.69 | 65 | 0.04 |
| 643.8 | 93.9 | 6.1 | Cal 3 | 24 | 5 | 20 | 334 | 2.61 | 0.80 | 0.53 | 0.14 | 0.68 | 0.16 | 0.04 | 0.21 | 0.03 | 0.18 | 0.03 | 0.09 | 0.01 | 0.08 | 0.01 | 0.003 | 5.61 | 0.61 | 0.53 | 77 | 0.01 |
| 683.8 | 100 | 0 | Cal 3 | 61 | 21 | 104 | 707 | 3.26 | 0.88 | 0.57 | 0.13 | 0.66 | 0.15 | 0.04 | 0.22 | 0.03 | 0.19 | 0.04 | 0.11 | 0.01 | 0.10 | 0.01 | 0.01 | 6.39 | 0.64 | 0.43 | 86 | 0.03 |
| 716.4 | 100 | 0 | Cal 3 | 168 | 50 | 187 | 661 | 2.89 | 0.88 | 0.83 | 0.17 | 0.81 | 0.19 | 0.05 | 0.25 | 0.03 | 0.22 | 0.04 | 0.12 | 0.01 | 0.11 | 0.01 | 0.04 | 6.62 | 0.72 | 0.49 | 70 | 0.08 |
| 759.9 | 1 | 98 | Dol 6 | 49 | 6 | 34 | 176 | 1.78 | 0.68 | 0.48 | 0.12 | 0.58 | 0.13 | 0.03 | 0.18 | 0.02 | 0.15 | 0.03 | 0.07 | 0.01 | 0.06 | 0.01 | 0.02 | 4.34 | 0.61 | 0.66 | 66 | 0.03 |
| 767.7 | 2 | 97 | Dol 6 | 22 | 6 | 13 | 179 | 1.39 | 0.57 | 0.35 | 0.09 | 0.46 | 0.11 | 0.03 | 0.14 | 0.02 | 0.12 | 0.02 | 0.06 | 0.01 | 0.05 | 0.005 | 0.01 | 3.42 | 0.56 | 0.61 | 65 | 0.03 |
| 775.1 | 2 | 96 | Dol 6 | 35 | 10 | 28 | 184 | 2.89 | 1.10 | 0.68 | 0.19 | 0.93 | 0.21 | 0.05 | 0.28 | 0.04 | 0.23 | 0.04 | 0.11 | 0.01 | 0.09 | 0.01 | 0.02 | 6.85 | 0.57 | 0.69 | 70 | 0.06 |
| 824.8 | 99 | 0 | Cal 4 | 31 | 11 | 38 | 656 | 3.98 | 1.06 | 0.78 | 0.18 | 0.88 | 0.20 | 0.05 | 0.27 | 0.04 | 0.25 | 0.05 | 0.14 | 0.02 | 0.11 | 0.01 | 0.01 | 8.02 | 0.63 | 0.52 | 81 | 0.02 |
| 856.8 | 98 | 2 | Cal 4 | 5 | 8 | 14 | 352 | 4.34 | 0.93 | 0.55 | 0.16 | 0.81 | 0.19 | 0.05 | 0.28 | 0.04 | 0.27 | 0.05 | 0.16 | 0.02 | 0.13 | 0.01 | 0.01 | 8.00 | 0.52 | 0.39 | 81 | 0.02 |
| 896.7 | 1 | 98 | — | 26 | 6 | 14 | 159 | 1.92 | 0.55 | 0.39 | 0.10 | 0.49 | 0.12 | 0.03 | 0.16 | 0.02 | 0.15 | 0.03 | 0.08 | 0.01 | 0.07 | 0.01 | 0.02 | 4.14 | 0.58 | 0.43 | 68 | 0.03 |
| 934.8 | 95 | 4 | — | 10 | 11 | 29 | 461 | 3.93 | 0.86 | 0.92 | 0.15 | 0.75 | 0.18 | 0.05 | 0.27 | 0.04 | 0.26 | 0.05 | 0.15 | 0.02 | 0.13 | 0.01 | 0.01 | 7.78 | 0.95 | 0.35 | 77 | 0.02 |
| 973.8 | 4 | 96 | Dol 7 | 43 | 9 | 36 | 171 | 2.28 | 0.67 | 0.48 | 0.12 | 0.60 | 0.15 | 0.04 | 0.20 | 0.03 | 0.18 | 0.03 | 0.10 | 0.01 | 0.09 | 0.01 | 0.02 | 5.00 | 0.59 | 0.44 | 67 | 0.05 |
| 997.1 | 5 | 95 | Dol 7 | 21 | 10 | 12 | 184 | 2.46 | 0.71 | 0.44 | 0.13 | 0.63 | 0.16 | 0.04 | 0.21 | 0.03 | 0.19 | 0.04 | 0.10 | 0.01 | 0.10 | 0.01 | 0.02 | 5.25 | 0.53 | 0.42 | 69 | 0.05 |
| 1006.8 | 6 | 94 | Dol 7 | 72 | 11 | 52 | 204 | 2.19 | 0.73 | 0.53 | 0.13 | 0.64 | 0.15 | 0.04 | 0.20 | 0.03 | 0.18 | 0.03 | 0.10 | 0.01 | 0.09 | 0.01 | 0.03 | 5.05 | 0.61 | 0.49 | 68 | 0.06 |
| 1035.8 | 1 | 99 | Dol 7 | 71 | 12 | 84 | 171 | 2.31 | 0.86 | 0.86 | 0.16 | 0.78 | 0.19 | 0.04 | 0.23 | 0.03 | 0.20 | 0.04 | 0.11 | 0.01 | 0.09 | 0.01 | 0.05 | 5.92 | 0.79 | 0.55 | 63 | 0.07 |
| 1048.7 | 1 | 99 | Dol 7 | 89 | 16 | 97 | 203 | 4.46 | 1.16 | 1.78 | 0.28 | 1.37 | 0.37 | 0.09 | 0.44 | 0.07 | 0.42 | 0.08 | 0.22 | 0.03 | 0.20 | 0.02 | 0.07 | 10.99 | 0.95 | 0.44 | 57 | 0.08 |

**Table 1.** *Cont.*

| Depth m | LMC % | DOL % | Mineral Unit | Al µg/g | Mn µg/g | Fe µg/g | Sr µg/g | Y µg/g | La µg/g | Ce µg/g | Pr µg/g | Nd µg/g | Sm µg/g | Eu µg/g | Gd µg/g | Tb µg/g | Dy µg/g | Ho µg/g | Er µg/g | Tm µg/g | Yb µg/g | Lu µg/g | Th µg/g | ∑REY µg/g | Ce/Ce* | (Pr/Yb)$_N$ | Y/Ho | Mn/Sr |
|---|---|---|---|---|---|---|---|---|---|---|---|---|---|---|---|---|---|---|---|---|---|---|---|---|---|---|---|---|
| 1064.6 | 1 | 99 | Dol 7 | 55 | 31 | 117 | 180 | 2.25 | 0.53 | 0.66 | 0.10 | 0.49 | 0.13 | 0.03 | 0.17 | 0.03 | 0.17 | 0.03 | 0.10 | 0.01 | 0.09 | 0.01 | 0.03 | 4.79 | 0.98 | 0.36 | 70 | 0.17 |
| 1078.4 | 1 | 99 | Dol 7 | 67 | 22 | 105 | 198 | 5.48 | 1.65 | 1.08 | 0.30 | 1.45 | 0.34 | 0.09 | 0.45 | 0.06 | 0.41 | 0.08 | 0.23 | 0.03 | 0.20 | 0.02 | 0.06 | 11.86 | 0.53 | 0.47 | 68 | 0.11 |
| 1094.8 | 1 | 99 | Dol 7 | 81 | 13 | 88 | 183 | 3.31 | 0.96 | 0.67 | 0.18 | 0.85 | 0.21 | 0.05 | 0.27 | 0.04 | 0.25 | 0.05 | 0.14 | 0.02 | 0.13 | 0.01 | 0.04 | 7.13 | 0.56 | 0.45 | 69 | 0.07 |
| 1102.8 | 1 | 99 | Dol 7 | 62 | 10 | 70 | 187 | 2.61 | 0.87 | 0.78 | 0.18 | 0.82 | 0.20 | 0.05 | 0.24 | 0.03 | 0.23 | 0.04 | 0.13 | 0.01 | 0.10 | 0.01 | 0.03 | 6.31 | 0.64 | 0.53 | 63 | 0.05 |
| 1112.0 | 1 | 99 | Dol 7 | 49 | 20 | 48 | 175 | 1.53 | 0.48 | 0.35 | 0.09 | 0.42 | 0.10 | 0.02 | 0.13 | 0.02 | 0.12 | 0.02 | 0.07 | 0.01 | 0.06 | 0.01 | 0.02 | 3.41 | 0.59 | 0.48 | 67 | 0.12 |
| 1122.0 | 1 | 99 | Dol 7 | 33 | 9 | 25 | 196 | 2.03 | 0.54 | 0.40 | 0.10 | 0.48 | 0.12 | 0.03 | 0.15 | 0.02 | 0.14 | 0.03 | 0.08 | 0.01 | 0.07 | 0.01 | 0.02 | 4.21 | 0.61 | 0.43 | 73 | 0.04 |
| 1137.0 | 1 | 99 | Dol 7 | 23 | 12 | 22 | 189 | 2.03 | 0.68 | 0.46 | 0.12 | 0.56 | 0.14 | 0.03 | 0.18 | 0.02 | 0.15 | 0.03 | 0.08 | 0.01 | 0.06 | 0.01 | 0.02 | 4.56 | 0.59 | 0.57 | 68 | 0.07 |
| 1147.0 | 1 | 99 | Dol 7 | 40 | 13 | 39 | 230 | 1.40 | 0.57 | 0.46 | 0.11 | 0.49 | 0.11 | 0.03 | 0.13 | 0.02 | 0.12 | 0.02 | 0.06 | 0.01 | 0.05 | 0.005 | 0.02 | 3.59 | 0.63 | 0.61 | 67 | 0.06 |
| 1152.0 | 1 | 99 | Dol 7 | 25 | 23 | 36 | 195 | 1.13 | 0.48 | 0.40 | 0.08 | 0.39 | 0.09 | 0.02 | 0.11 | 0.02 | 0.10 | 0.02 | 0.05 | 0.01 | 0.05 | 0.004 | 0.02 | 2.95 | 0.70 | 0.54 | 63 | 0.12 |
| 1167.0 | 1 | 99 | Dol 7 | 32 | 21 | 71 | 231 | 2.22 | 0.79 | 0.57 | 0.13 | 0.64 | 0.14 | 0.03 | 0.18 | 0.03 | 0.16 | 0.03 | 0.09 | 0.01 | 0.07 | 0.01 | 0.03 | 5.11 | 0.62 | 0.61 | 72 | 0.09 |
| 1179.5 | 4 | 96 | Dol 7 | 232 | 33 | 163 | 223 | 7.13 | 2.32 | 1.68 | 0.43 | 1.99 | 0.44 | 0.11 | 0.57 | 0.08 | 0.51 | 0.10 | 0.28 | 0.03 | 0.23 | 0.02 | 0.07 | 15.91 | 0.56 | 0.59 | 73 | 0.15 |
| 1196.0 | 96 | 4 | Cal 5 | 68 | 24 | 62 | 587 | 2.38 | 0.93 | 0.70 | 0.16 | 0.76 | 0.16 | 0.04 | 0.20 | 0.03 | 0.17 | 0.03 | 0.09 | 0.01 | 0.08 | 0.01 | 0.01 | 5.75 | 0.63 | 0.65 | 73 | 0.04 |
| 1249.7 | 95 | 5 | — | 110 | 75 | 150 | 281 | 2.66 | 0.95 | 1.19 | 0.20 | 0.91 | 0.22 | 0.06 | 0.27 | 0.04 | 0.25 | 0.05 | 0.13 | 0.02 | 0.12 | 0.01 | 0.08 | 7.08 | 0.85 | 0.51 | 59 | 0.27 |
| Average | | | | 66 | 16 | 66 | 404 | 2.59 | 0.78 | 0.63 | 0.14 | 0.70 | 0.17 | 0.04 | 0.22 | 0.03 | 0.20 | 0.04 | 0.11 | 0.01 | 0.10 | 0.01 | 0.03 | 5.77 | 0.66 | 0.48 | 66 | 0.06 |
| Average Calcite | | | | 89 | 28 | 90 | 768 | 3.12 | 0.82 | 0.72 | 0.15 | 0.74 | 0.18 | 0.04 | 0.23 | 0.03 | 0.22 | 0.04 | 0.13 | 0.01 | 0.12 | 0.01 | 0.03 | 6.57 | 0.68 | 0.42 | 73 | 0.05 |
| Average Dolomite | | | | 57 | 12 | 57 | 190 | 2.15 | 0.72 | 0.58 | 0.13 | 0.64 | 0.15 | 0.04 | 0.19 | 0.03 | 0.18 | 0.03 | 0.10 | 0.01 | 0.08 | 0.01 | 0.03 | 5.04 | 0.65 | 0.51 | 64 | 0.06 |

Note: (1) ARA: aragonite, HMC: high-magnesium calcite, LMC: low-magnesium calcite, DOL: dolomite. (2) Cal 1: calcite unit 1, Dol 1: dolomite unit 1. "—" denotes non-mineral unit samples. (3) Ce/Ce* and (Pr/Yb)$_N$ values are normalized to Post-Archean Australian Shale (PAAS) [42].

*4.2. Diagenetic-Alteration-Sensitive Element Geochemical Proxies*

In the samples, the Al content ranges from 5 to 359 µg/g, with an average of 66 µg/g. The Th content varies from 0.003 to 0.08 µg/g, with an average of 0.03 µg/g. The Mn content ranges from 1 to 75 µg/g, with a mean value of 16 µg/g. The Fe content is from 3 to 293 µg/g, with a mean value of 66 µg/g. The Sr concentration varies between 159 and 3231 µg/g, with a mean value of 404 µg/g. The Mn/Sr ratio is between 0.004 to 0.27, averaging 0.06.

The Sr concentration is closely related to the mineralogical composition: higher than 2000 µg/g in the surficial sample containing ARA, between 200–2000 µg/g in pure calcite samples, and mostly less than 200 µg/g in pure dolomite samples. The other proxies show low values in both calcite and dolomite samples. This may suggest negligible effects from terrigenous materials and post-depositional diagenetic alteration (see Section 5.2 for details).

*4.3. REY Characteristics*

In calcite samples, $\sum$REY ranges from 2.79 to 11.28 µg/g, with an average of 6.57 µg/g. Ce/Ce* varies from 0.52 to 0.95, with an average of 0.68. $(Pr/Yb)_N$ ranges from 0.22 to 0.65, with an average of 0.42. Y/Ho ratio is from 59 to 86, with an average of 73. In dolomite samples, $\sum$REY ranges from 1.47 to 15.91 µg/g, with an average of 5.04 µg/g. Ce/Ce* varies from 0.51 to 1.14, with an average of 0.65. $(Pr/Yb)_N$ ranges from 0.35 to 0.72, with an average of 0.51. Y/Ho ratio is from 46 to 73, with an average of 64.

Overall, ranges and mean values of $\sum$REY, Ce/Ce*, $(Pr/Yb)_N$, and Y/Ho are all similar for calcite and dolomite samples. This may indicate that the mineral transformation from calcite to dolomite does not necessarily result in distinguishable changes in the REY composition of carbonates. Furthermore, the REY proxies (low $\sum$REY, Ce/Ce* < 1, $(Pr/Yb)_N$ < 1, and Y/Ho > 44) show typical marine values (see Sections 5.3 and 5.4 for detailed discussion).

## 5. Discussion

*5.1. Diagenesis Types*

Based on the mineralogical composition, petrographic microscopic features, and carbon and oxygen isotopic data, the XK-1 core carbonates are identified to have experienced three types of diagenesis: meteoric diagenesis, marine burial diagenesis, and dolomitization.

Meteoric diagenesis is identified in the strata above 180 m. Meteoric diagenesis refers to the cementation and lithification of unconsolidated carbonate sediments into carbonate rocks when exposed to atmospheric freshwater in the vadose and phreatic zones [48]. In this interval (depths of 0–180 m), the XK-1 core is composed of unconsolidated and semi-consolidated limestones (Figure 3a). Typical diagenetic features including dissolution, cementation, and neomorphism are observed on petrographic micrographs (Figure 3e–h). Cements around grains are typically present as meniscus (Figure 3e) and pore-filling isopachous dogtooth fringe (Figure 3g). Dissolved pores and residual intragranular rounding pores are well-developed (Figure 3e–h). These petrographic features are typical products of meteoric diagenesis [48,49].

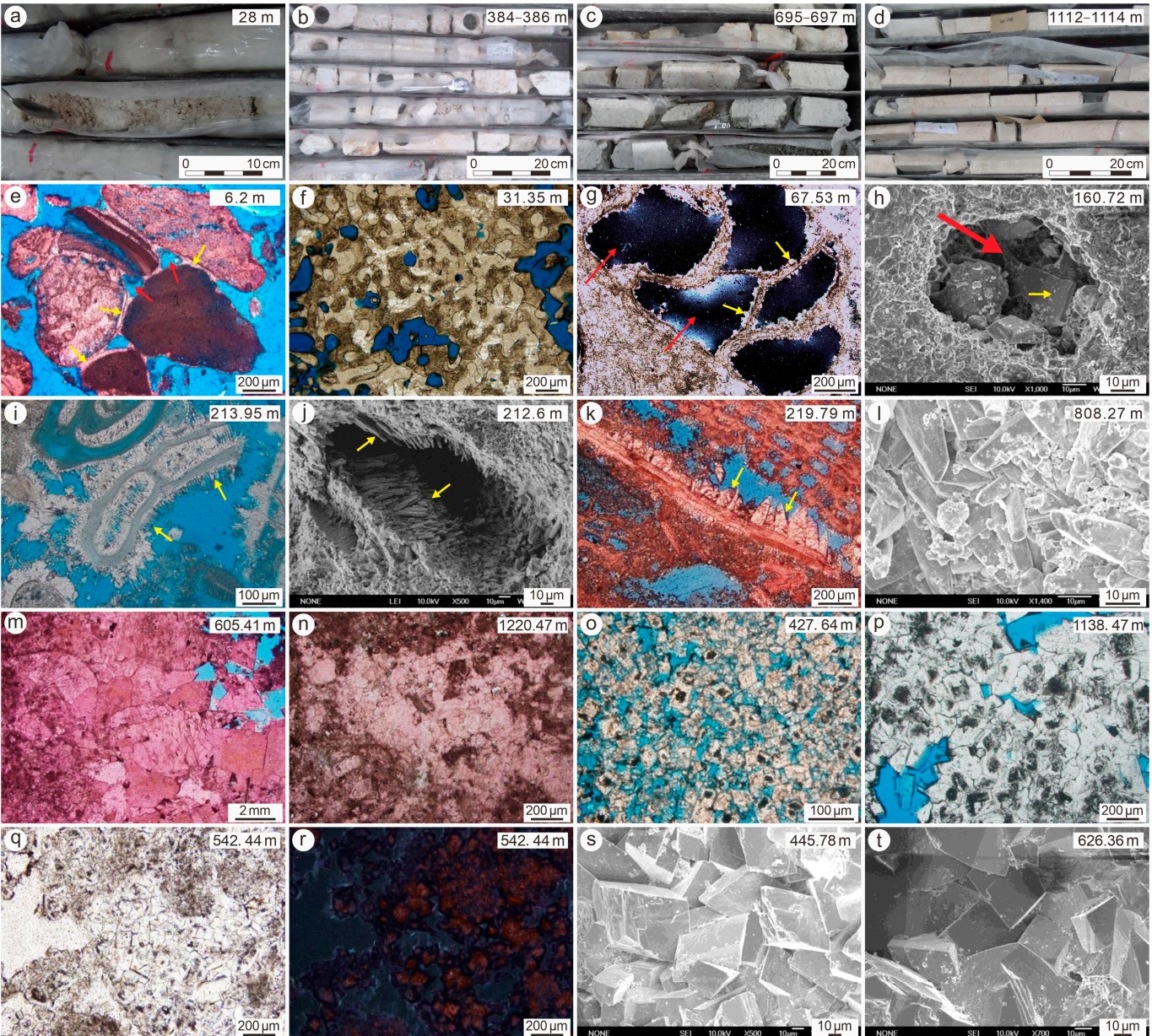

**Figure 3.** Photographs of core hand specimens and petrographic micrographs showing diagenetic features of the XK-1 core carbonates. (**a**–**d**) Photographs of core hand specimens at different depths (28 m, 384–386 m, 695–697 m, 1112–1114 m). (**e**–**h**) Typical products of meteoric diagenesis. (**e**) Meniscus calcite cements (yellow arrows), and rounded pores (red arrows), 6.2 m, plane-polarized light, thin section stained with blue epoxy. (**f**) Neomorphism in framestone, 31.35 m, plane-polarized light, stained section. (**g**) Melodic pores (red arrows) formed by dissolution, and pore-filling isopachous dogtooth fringes (yellow arrows), 67.53 m, plane-polarized light, stained section. (**h**) Melodic pores (red arrows) formed by dissolution, filled with fine granular rhombic calcite cements (yellow arrows), 160.72 m, scanning electron micrographs (SEM photographs). (**i**–**l**) Typical products formed under marine conditions. (**i**) Aragonite needle cements (yellow arrows) around bioclastics in packstone, 213.95 m, plane-polarized light, stained section. (**j**) Circumgranular fibrous and bladed calcite marine cements (yellow arrows) in wackestone, 212.6 m, SEM photographs. (**k**) Bladed calcite cements (yellow arrows) in packstone, 219.79 m, plane-polarized light, stained section. (**l**) Pore-filling bladed calcite marine cements, 808.27 m, SEM photographs. (**m,n**) Typical signs of burial diagenesis.

(**m**) Coarse poikilotopic blocky calcite spar, 605.41 m, plane-polarized light, stained section. (**n**) Coarse poikilotopic blocky calcite spar, 1220.47 m, plane-polarized light, stained section. (**o–t**) Typical products of dolomitization under stable marine conditions. (**o**) Dolomite rhombs with cloudy cores and clear rims, 427.64 m, plane-polarized light, stained section. (**p**) Dolomite rhombs with smaller cloudy cores and larger bright exteriors, and extensive interlocking of adjacent crystals by cementation, 1138.47 m, plane-polarized light, stained section. (**q**) Dolomite rhombs with cloudy cores and clear rims, 542.44 m, plane-polarized light. (**r**) Cathodoluminescence (CL) image of the previous photograph shows orange cores and dark rims, and the replacement and cementation relationships are also shown. (**s**) Euhedral dolomite rhombs, 445.78 m, SEM photographs. (**t**) Euhedral dolomite rhombs, 626.36 m, SEM photographs.

Based on mineralogical composition, three sub-intervals are further identified. In the top sub-interval (0–22 m), the core is composed of LMC, HMC, and ARA (Figure 2). The $\delta^{13}$C values of the core samples are slightly positive and the $\delta^{18}$O values are slightly negative (Figures 2 and 4). The presence of primary ARA and HMC, together with the $\delta^{13}$C and $\delta^{18}$O signatures, suggests that this sub-interval is only slightly altered by meteoric fluids. In the middle sub-interval (22–36 m), the core consists mainly of LMC and ARA. According to the magnetostratigraphic and $^{230}$Th dating, the corresponding age of the XK-1 core at 36 m depth is less than 0.2 Ma [50]. The transformation from HMC to LMC induced by freshwater can be completed in a few thousand years, whereas calcification of ARA may take tens of thousands of years [51]. Therefore, the preservation of ARA and the disappearance of HMC may indicate a further alteration by meteoric diagenesis. This indication can also be supported by changes in the carbon and oxygen isotopic compositions. The $\delta^{13}$C values of the core samples become negative and the $\delta^{18}$O values decrease abruptly to less than −8‰. The infiltration and circulation of meteoric freshwater enriched in $^{12}$C and $^{16}$O could result in negative excursions of both $\delta^{13}$C and $\delta^{18}$O values in shallow water carbonates. When the $\delta^{18}$O value is less than −5‰, and especially less than −10‰, the primary carbonate minerals are indicated to have been significantly altered by meteoric fluids [52,53]. In the bottom sub-interval (36–180 m), the core is composed exclusively of LMC. The unstable primary ARA and HMC have all transformed into stable LMC. The $\delta^{13}$C values become more negative, with a minimum close to −6‰, and the $\delta^{18}$O values are consistently as low as approximately −8‰. These features demonstrate complete meteoric diagenetic alteration. In addition, there are five exposure surfaces within the meteoric diagenesis zone (0–2.9 m, 21.93–22.41 m, 37.3–38.15 m, 68.67–72.07 m, and 97.58–98.84 m) [44,47], which undoubtedly facilitate the infiltration of freshwater from the surface, leading to carbonate alteration.

All carbonate strata below 180 m have experienced varying degrees of marine burial diagenesis. Marine burial diagenesis refers to shallow burial diagenesis that occurs in the marine environment, which influences at depths between the meteoric diagenesis zone and the deeper burial diagenesis zone [6,49,54–56]. During marine burial diagenesis, carbonate cementation, recrystallization, and polymorphic transformations occur prevalently under the influence of marine pore fluids [49]. In the discussion of marine burial diagenesis here, in order to exclude the superimposed effects of dolomitization, only limestone intervals exclusive of strong overprints by dolomitization are involved. Within this interval (180–1257.52 m), all primary minerals (ARA and HMC) have been stabilized as LMC (Figure 2). The fibrous ARA is present sporadically, as a product of cementation under marine conditions (Figure 3i). Below the depth of 565 m (top boundary of calcite unit 2), coarse poikilotopic blocky calcite spars begin to appear (Figure 3m), which is a sign of elevated burial diagenesis [57]. In the bottom of the core, they are interlocked tightly, leaving almost no pores (Figure 3n). This may be a result of cementation promoted by further burial diagenesis. However, the imprints of earlier marine diagenesis are widely preserved. For example, at 808.27 m, bladed calcites are still well present (Figure 3l), which were formed in the marine environment [58]. During burial, the $\delta^{13}$C values of diagenetic carbonates typically do not change because the potential for oxidation of pore waters in the

diffusion regime is too low to affect the $\delta^{13}C$ values, while the $\delta^{18}O$ values decrease with depth as a consequence of carbonate dissolution and reprecipitation under the influence of increasing burial temperature [59]. Although there exist widespread cements as the product of dissolution and reprecipitation in the burial zone, the $\delta^{13}C$ and $\delta^{18}O$ values of the XK-1 core limestones do not show any decreasing trend with depth (Figure 2). This may indicate that the burial triggered the dissolution of primary carbonates in a relatively closed environment and generated secondary pore waters with similar geochemical composition to the primary values. The limestones have a wide range of $\delta^{13}C$ and $\delta^{18}O$ values, some of which fall in the slightly altered zone, while the others overlap with the meteoric diagenesis zone (Figure 4). This may indicate that the $\delta^{13}C$ and $\delta^{18}O$ values in the samples before burial were inherited. Overall, the above features suggest a limited influence of marine burial diagenesis on the geochemical signatures of the XK-1 core carbonates.

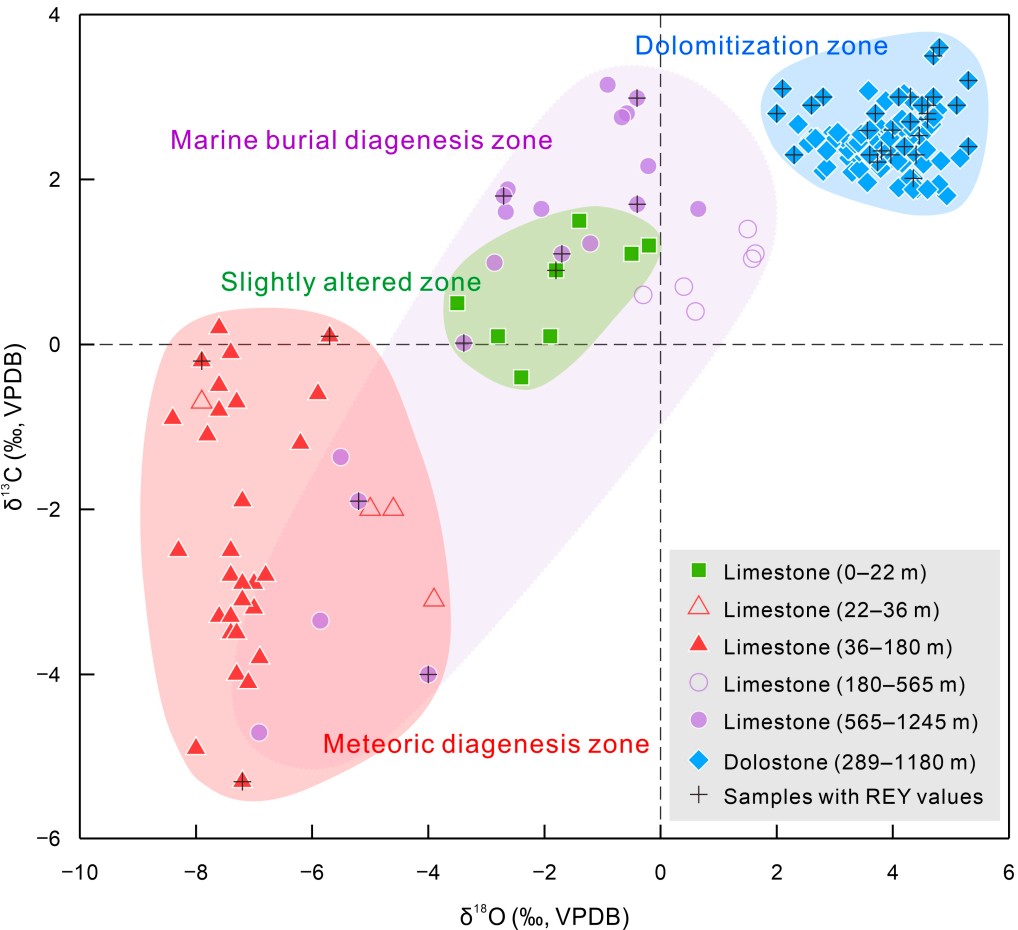

**Figure 4.** Cross plot of $\delta^{13}C$ and $\delta^{18}O$ values of the XK-1 core carbonates in different diagenesis zones. Diagenesis types, including meteoric diagenesis, marine burial diagenesis and dolomitization, are identified based on the petrographic features, mineralogical composition, and $\delta^{13}C$ and $\delta^{18}O$ values. $\delta^{13}C$ and $\delta^{18}O$ data on the VPDB scale are from [44–46]. The hollow symbols correspond to samples identified as experiencing weak diagenesis. The black cross symbols (+) represent samples with measured REY concentrations in this paper. Numbers in parentheses represent depth.

The dolomitization zone is composed of completely or partially dolomitized intervals (180–565 m, 615–637 m, 758–780 m, and 850–1184 m, Figure 2). The petrological hand specimens show that the rocks in these intervals were further compacted (Figure 3d). Dolomite rhombs with cloudy cores and clear rims are ubiquitously observed throughout the dolomitized intervals (Figure 3o–q). In the CL image, the cores are shown as orange and the rims are less luminescent (Figure 3r), which is an extremely common characteristic

in marine dolomites. The lighter-luminescing cores have been interpreted as heritages of metastable dolomites formed earlier in marine-meteoric mixing zone conditions, and the darker-luminescing rims as replacement dolomites produced later in more marine conditions [60,61]. The dolomite crystals in deeper strata (e.g., dolomite unit 7, Figure 3o) are larger, with smaller cloudy centers and larger bright exteriors, than those in shallower strata (e.g., dolomite unit 4, Figure 3p), indicating an increasing degree of recrystallization with depth. This coarsely crystalline dolomite also shows extensive interlocking of adjacent crystals by cementation (Figure 3p), suggesting that dolomitization substantially indurated the rock and reduced porosity. SEM photographs reveal that dolomite crystals are largely euhedral (Figure 3s,t), which may reflect slow crystallization under stable marine conditions. The dolomite samples have positive $\delta^{13}C$ (>1.5‰) and $\delta^{18}O$ (>2‰) values (Figures 2 and 3) similar to the reported values of Bahamian dolomites, which have been interpreted as a record of seawater signatures [62,63]. Recent studies [22,23,46] suggested that the Xisha dolomite (dolomite in the Xisha Islands) forms in the near-surface low-temperature environment, the dolomitizing fluid is chemically similar to the seawater-sourced hypersaline brine, and the possible mechanism for dolomitization has been proposed to be a seepage–reflux dolomitization model [64–66]. Regionally, Late Miocene dolomite intervals about 200 m thick have been uniformly found in wells XK-1, XY-1, XY-2, XC-1, and CK-2 (Figure 1b). This may indicate that the dolomitization of the Xisha carbonates is at least a regional event rather than a local event. Apart from discovery in the Xisha area, Neogene island dolomites are also widely distributed all over the world, indicating that their formation may be controlled by globally unified paleoclimate and paleo-oceanographic conditions [14,16,67–69].

In summary, all three most common diagenesis types exist at different depths of the Xisha carbonate platform: meteoric diagenesis, marine burial diagenesis, and dolomitization. This makes the Xisha carbonates ideal for evaluating the retention of original seawater REY characteristics after varying diagenesis. Specifically, shallow calcite samples (located in calcite unit 1, depths of 35–180 m) can be used to assess the influence of meteoric diagenesis on the original seawater REY signatures preserved in carbonates, deep calcite samples (calcite units 2–5, 580–1211 m) and dolomite samples (dolomite units 1–7, 289–1180 m) can be used to assess the influence of marine burial diagenesis and dolomitization, respectively.

### 5.2. Effects of Diagenetic Alteration on the REY Signatures

Before evaluating the effects of diagenesis on carbonate rocks, the first step is to eliminate the contribution of terrigenous detritus [12,13]. As we have stated in the methods section, we extracted carbonate components by a chemical leaching protocol before measuring their elemental concentrations. Al and Th in marine carbonates are common tracers of terrigenous detritus [12,13]. The Al contents of the XK-1 core samples are not higher than 360 µg/g, and mostly lower than 100 µg/g, while the Th contents of all samples are less than 0.1 µg/g (Figure 2). The low contents of Al and Th indicate that the contribution of siliciclastic detritus is negligible. In addition, there are no correlations between the Al content and the values of the REY proxies ($\sum$REY, Ce/Ce*, (Pr/Yb)$_N$, and Y/Ho) (Figure 5), indicating that the effects of terrigenous input on the REY composition of samples are negligible. Finally, the low REY concentrations in the samples (mostly lower than 20 µg/g, Table 1 and Figure 2) also exhibit a typical marine signature without contamination from terrigenous detritus.

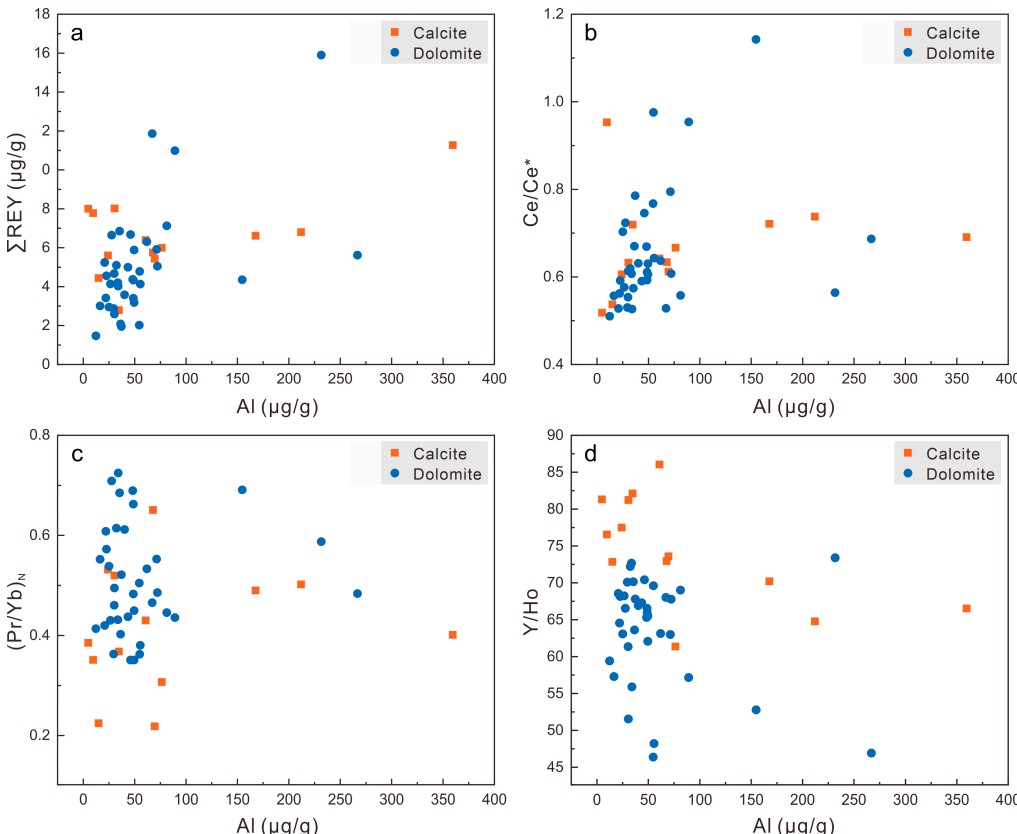

**Figure 5.** (**a**) ∑REY, (**b**) Ce/Ce*, (**c**) (Pr/Yb)$_N$, and (**d**) Y/Ho vs. the Al concentration in the XK-1 core carbonates. For both calcite and dolomite samples, the Al concentration does not show any correlations with the values of REY proxies.

The contents of Fe and Mn in all samples are relatively low, mostly lower than 300 µg/g and 100 µg/g, respectively. The contents of Fe and Mn in dolomite samples are generally lower than 100 µg/g and 30 µg/g, respectively (Figure 2). Deep hydrothermal fluids are enriched in Fe and Mn due to the influence of basic rocks or magmatic materials, and Fe and Mn prevail in atmospheric particulates and surface runoff [70]. The contents of Fe and Mn in the samples are much lower than the lower limit values of diagenetic alteration [6,71]. This indicates that seawater is the major source of Fe and Mn in the samples and the influence of potential post-depositional diagenetic fluids (such as the deep hydrothermal input or circulation of freshwater in shallow carbonate strata) is minimal. In the XK-1 core carbonate samples, the Sr content in aragonite, calcite, and dolomite decreases gradually (Figure 2). The high Sr content is mainly distributed above 180 m throughout the core, where coral fossils are abundant (an average of 50% in all species) [33,35], which may reflect the enrichment of Sr by biological activities. Most deep or surficial fluids are characterized by high Mn and low Sr content [70]; thus, the mineral recrystallization, neomorphism, or dolomitization induced by these fluids could increase the Mn/Sr ratios of secondary carbonates. The Mn/Sr ratios of all samples are less than 0.3 (Figure 2), which is much lower than the commonly used diagenetic alteration threshold (Mn/Sr = 1–2) [72], indicating that the XK-1 core carbonates experienced limited diagenetic alteration. Further, other diagenetic alteration sensitive proxies (Mn/Sr and δ$^{18}$O) show no correlations with either ∑REY or Ce/Ce* (Figure 6), confirming the limited influence of diagenesis on the REY composition of samples.

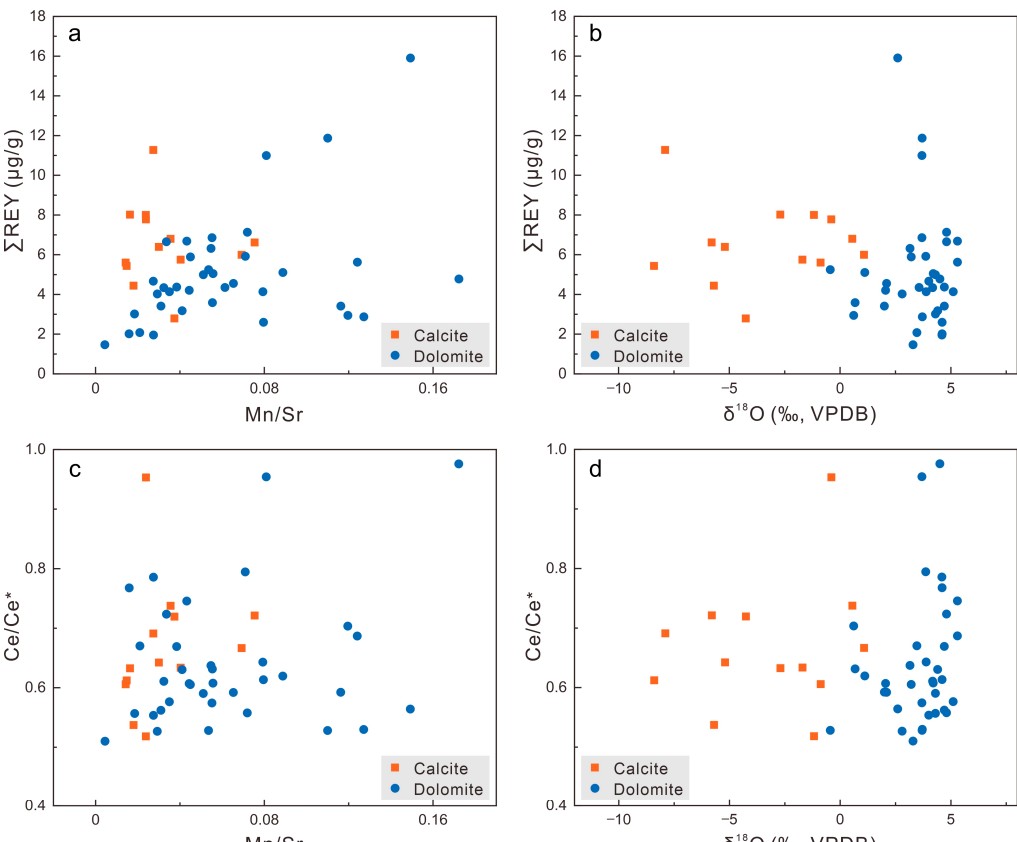

**Figure 6.** (**a**) ∑REY vs. Mn/Sr, (**b**) ∑REY vs. δ¹⁸O, (**c**) Ce/Ce* vs. Mn/Sr, (**d**) Ce/Ce* vs. δ¹⁸O in the XK-1 core carbonates. The δ¹⁸O data are from [44–46]. For both calcite and dolomite samples, neither ∑REY nor Ce/Ce* show any correlations with Mn/Sr or δ¹⁸O.

In summary, even under the influence of meteoric diagenesis, marine burial diagenesis, or dolomitization, the original REY signatures preserved in the Xisha carbonates have not been significantly altered by later fluids in various post-depositional processes.

*5.3. Potential of the Xisha Carbonates for Preserving Original Seawater REY Signatures*

The average REY contents of carbonate samples from both calcite and dolomite units in the XK-1 core are calculated. Using these average values, we plot the PAAS-normalized REY distribution patterns in Figure 7. The average REY value for the upper 100 m of modern seawater in the South China Sea [73] has also been normalized against PAAS for comparison. The main reasons for choosing the average shallow seawater REY value are: (1) island carbonates are formed in shallow seawater; (2) seawater REY values vary greatly with longitudinal distribution but are relatively uniform within the upper 100 m of the South China Sea [73]. We find that the REY patterns of both five calcite units (Figure 7a) and seven dolomite units (Figure 7b) in the XK-1 core are similar to the average seawater REY pattern, uniformly characterized by LREE depletion relative to HREE, negative Ce anomalies, and enrichments of Y relative to Ho. Due to the lanthanide contraction effect and the higher complexation ability of LREE with carbonate ions, LREE in seawater is deficient relative to HREE, which can be quantified as $(Pr/Yb)_N < 1$ [2]. The average calculated $(Pr/Yb)_N$ values in the calcite and dolomite samples are 0.42 and 0.51, respectively, consistent with normal seawater values. In oxic seawater, $Ce^{3+}$ is easily oxidized to insoluble $Ce^{4+}$ and precipitated from seawater, which could result in the depletion of Ce relative to adjacent elements and a negative anomaly of Ce in seawater [74]. The mean Ce/Ce* values are 0.68 and 0.65 in the calcite and dolomite samples, displaying typical oxic seawater signatures. Owing to the weaker surface complexation capacity, Y is less prone to be scavenged from seawater compared with Ho, thus producing a high Y/Ho ratio in seawater (>44) [75]. Typical Y/Ho

ratios in marine carbonates are in the range of 44–74 [75]. The average Y/Ho ratios in the calcite and dolomite samples are 73 and 64, respectively, which fall in the typical value range of marine carbonates. There is no distinguishable REY pattern difference between calcite and dolomite (Figure 7c). The average samples in the meteoric diagenesis zone (calcite unit 1), marine burial diagenesis zone (calcite units 2–5), and dolomitization zone (dolomite units 1–7) exhibit similar REY patterns (Figure 7d). In summary, regardless of mineralogical composition and diagenetic overprints, both calcite and dolomite samples have faithfully preserved the original REY signals of seawater.

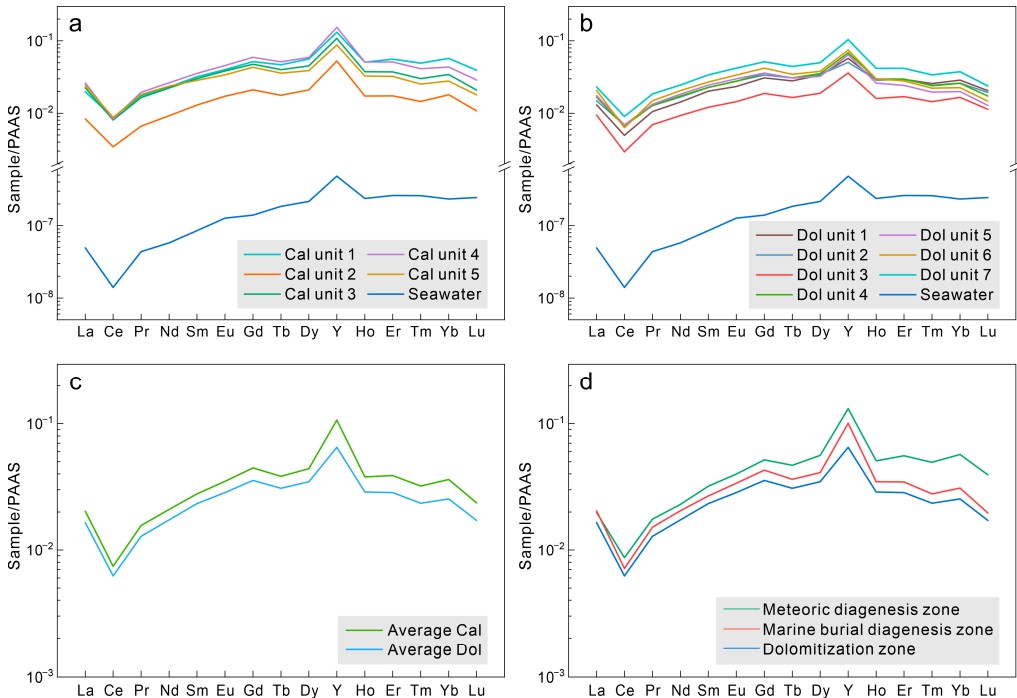

**Figure 7.** (**a**) PAAS-normalized REY distribution patterns of carbonate samples from the calcite units in the XK-1 core in comparison with modern seawater. (**b**) PAAS-normalized REY distribution patterns of the dolomite unit samples. (**c**) Comparison of PAAS-normalized REY patterns in average calcite with average dolomite. (**d**) Average REY patterns comparison in different diagenesis zones of the core carbonates. Cal: calcite, Dol: dolomite. Seawater value is calculated from the average REY value of 100 m deep modern seawater in the South China Sea [73].

Our study shows calcite and dolomite are able to preserve seawater REY signatures without being affected by various diagenesis. During early marine diagenesis, the preservation of seawater REY signatures in calcite could be attributed to the influence of seawater-dominated diagenetic fluids. In the post-depositional process, neither meteoric diagenesis nor marine burial diagenesis can significantly modify the primary REY composition retained in calcite. This can be explained through two aspects. The first is the geochemical properties of the diagenetic fluid. As the REY abundance in freshwater is generally low, it will be difficult to change the original REY composition of primary carbonate sediments if the meteoric diagenetic alteration does not last for a long time [76–80]. Although exposure surfaces occurred frequently in the XK-1 core, most of them are thin. For example, the thicknesses of the five exposure surfaces above 100 m are 2.9 m, 0.48 m, 0.85 m, 3.4 m, and 1.26 m [44,47]. This suggests that the exposed periods of carbonate sediments are relatively short. Therefore, meteoric diagenesis does not necessarily cause significant alteration to the original REY composition in calcite samples. In contrast to freshwater, deep pore waters are usually enriched in REY [81]. Nevertheless, to significantly modify the REY composition of primary carbonates, pore waters must have very different REY characteristics and be present in large quantities (e.g., high pore water to rock ratios). In general, marine burial

diagenesis may cause the dissolution of some primary carbonate minerals, which might be the major source of REY to deep pore waters [6]. This means that deep pore waters may have REY composition similar to that of primary carbonates, and the primary REY signals will remain largely unchanged during marine burial diagenesis. This can be supported by the preservation of marine bladed calcite cements in the XK-1 core samples despite the burial process (Figure 3j–l). The second aspect is the composition and structure of the rock. As we have discussed in Section 5.1, the cementation during burial further cemented and interlocked the coarse poikilotopic calcite crystals (Figure 3m,n), decreasing the porosity and permeability of carbonate rocks. This could foster a relatively closed environment, reduce the contact between pore waters and island carbonates, and weaken the ability of REY in pore waters to enter the crystal lattice of minerals [82]. The above understanding can also be supported by the REY geochemical studies of ancient carbonates. For example, although the late Devonian reefal carbonates in the Canning basin of Western Australia recrystallized under the influence of diagenetic fluids, most limestones could still retain the original REY characteristics of the seawater in which they precipitated [2]. The early Archean stromatolite reefal carbonates in the Pilbara craton of Western Australia also show the Archean marine REY patterns [83]. These cases suggest that ancient carbonates are able to preserve the REY signatures of the waters in which they precipitated. These studies and our results from the calcite samples in the XK-1 core collectively suggest that secondary carbonates still have the potential to retain primary seawater REY signatures, regardless of mineralogical transformations during meteoric diagenesis or marine burial diagenesis.

The retention of seawater REY signals in dolomite may reflect another situation; that is, the dolomitizing fluid has REY geochemical composition similar to that of contemporaneous seawater. Previous studies have demonstrated that the dolomitization in the Xisha carbonate platform occurred in near-surface shallow marine environments through a seepage–reflux model where seawater-derived hypersaline waters were the dolomitizing fluids [22,23,46]. The stratigraphic age of the Xisha carbonates constrained by Sr isotopes is consistent with the paleomagnetic and biostratigraphic age [24,34], suggesting the dolomitizing fluids have the same $^{87}Sr/^{86}Sr$ ratio as contemporaneous seawater. This indicates that the dolomitization is penecontemporaneous, which may have occurred shortly after the deposition of primary carbonates. This could also be well supported by petrographic features. Largely euhedral dolomite rhombs with cloudy cores and clear rims (Figure 3o–t) reflect relatively slow replacement and recrystallization under stable marine conditions. By transforming precursor minerals (aragonite or calcite) to dolomite, dolomitization may completely reset the original geochemical composition preserved in primary carbonates. However, since the chemical composition of the dolomitizing fluids is similar to that of contemporaneous seawater, this resetting effect could reinforce the preservation of seawater REY signals in dolomite. In the post-depositional diagenetic process, dolomite is more stable and more resistant to dissolution than calcite at a wide range of temperatures and pressures [84]. Petrographic features show that the deep burial process could form cements to indurate dolomites and reduce porosity, but it hardly leads to complete dissolution of dolomites (Figure 3p). Therefore, the original seawater REY signals preserved in dolomite during dolomitization are rather resistant to later modifications by pore fluids. The above understanding also explains why ancient dolostone could be a good archive of the geochemical composition of ancient seawater. For example, even if subject to complete dolomitization, well-preserved Carboniferous dolostones may still have the REY characteristics of contemporaneous seawater [11]. To sum up, the dolomitization induced in the marine environment would not destroy but rather promote the preservation of the original seawater REY signatures in carbonates.

### 5.4. Implications of Using the Ce Anomaly in Marine Carbonates as a Paleo-Redox Tracer

The Ce anomaly is commonly used to fingerprint oxic versus anoxic depositional conditions [1]. Under oxic marine conditions, Ce is preferentially adsorbed by manganese and iron oxides and hydroxides and removed from the water column [85], leading to a

negative anomaly (Ce/Ce* < 1). Under anoxic conditions, reductive dissolution of the Mn- and Fe-rich fractions can occur, resulting in a weak or even absent Ce anomaly (Ce/Ce* close to 1) [86]. Similarly, Ce/Ce* is close to or even greater than 1 during post-depositional diagenetic alteration when influenced by diagenetic fluids with the contribution of Mn-Fe oxide dissolution. Thus, Ce/Ce* in carbonate rocks insignificantly affected by Mn-Fe oxides can be used to distinguish between oxic and anoxic depositional environments [87].

The average value of Ce/Ce* in dolomite unit samples of the XK-1 core is consistent with that in calcite unit samples. This indicates that the Ce/Ce* values in carbonates do not change significantly, although the mineral structure and composition have been changed during the transformation from calcite to dolomite. Mn and Fe contents in the samples below the lower limit of diagenetic alteration preclude the influence of Mn-Fe oxides on Ce/Ce* values [6,71]. No correlations between Ce/Ce* and typical diagenetic-alteration-sensitive proxies (Mn/Sr and $\delta^{18}O$) (Figure 6c,d) indicate that the Ce/Ce* values in carbonates remain largely unchanged during multiple types of diagenesis, including meteoric diagenesis, marine burial diagenesis, and dolomitization. The relatively low and uniform Ce/Ce* values (average of 0.66) in the XK-1 core (Figure 2) indicate stable oxic ocean conditions in the South China Sea since the Neogene. This conclusion is supported by the presence of macrofossils such as corals, echinoderms, and brachiopods in the XK-1 core [33–35] and the characteristics of redox-sensitive element proxies [88]. Therefore, the Ce/Ce* values in carbonates are not controlled by either diagenesis type or mineralogical composition. Rather, they directly reflect the redox conditions of the ambient seawater during primary carbonate deposition.

This research supported the efforts of using the Ce anomaly in shallow marine carbonates to trace the redox evolution of Earth's near-surface environments over geologic history [89,90]. Carbonates, especially dolomites, could provide a continuous record back to the Proterozoic ocean.

## 6. Conclusions

The main conclusions are as follows.

(1) The reefal carbonates in the Xisha Islands of the South China Sea well record the original seawater REY signatures during primary deposition. Meteoric diagenesis, marine burial diagenesis, and dolomitization do not drive the REY patterns and the values of typical REY proxies (Ce/Ce*, Y/Ho, and $(Pr/Yb)_N$) away from primary seawater signatures.

(2) The Ce anomaly in diagenetically altered shallow marine carbonates can still be used as a good proxy for the redox conditions of the surrounding waters during primary carbonate deposition.

(3) The Ce/Ce* characteristics indicate that water column conditions for the formation of the Xisha carbonates have been constantly oxic from the Neogene to the present, consistent with the conclusion inferred from paleontological fossils and redox-sensitive elemental proxies.

**Author Contributions:** Conceptualization, X.-F.L. and X.-M.L.; methodology, X.-F.L., X.-K.W. and X.-M.L.; investigation, X.-F.L. and X.-K.W.; resources, X.L., X.-K.W. and X.-M.L.; data curation, X.-M.L.; writing—original draft preparation, X.-F.L.; writing—review and editing, S.Z., X.-M.L. and X.-F.L.; visualization, X.-F.L. and X.-K.W.; supervision, S.Z. and X.-M.L.; project administration, S.Z.; funding acquisition, S.Z. All authors have read and agreed to the published version of the manuscript.

**Funding:** This research was funded by the University of North Carolina at Chapel Hill, the Project of China National Offshore Oil Corporation (CNOOC) Limited under contract No. CCL2013ZJFN0729, the National Science and Technology Major Project under contract No. 2011ZX05025-002-03, and the Overseas Joint Training Project for Doctoral Students of Ocean University of China.

**Data Availability Statement:** Not applicable.

**Acknowledgments:** We would like to thank the editors and three anonymous reviewers for their valuable comments, which helped to greatly improve the manuscript.

**Conflicts of Interest:** The authors declare no conflict of interest.

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
