# Peer review of "Rare Earth Element Geochemistry of Late Cenozoic Island Carbonates in the South China Sea"

_minerals, doi:10.3390/min12050578_

Round 1
Reviewer 1 Report
Dear Authors, your ms deals with an interesting topic that deserves to be explored. I think your paper is well written and organized although some sentences have to be revised to better explain your considerations. More attention must to be paid to the data presentation that is not a repetition of a table. Maybe the addition of descriptive graphs can help you. However, my main concern about your paper is the lack of petrographic observations and/or detailed mineralogical data (i.e. EPMA, microdiffraction, WDS, ecc.) that are needed to support your inferences in the discussion and conclusion sections. You'll find details on my comments and suggestions in the attached pdf.
Accordingly, I suggest major revisions for this ms that in its present form is not suitable for publication.
Best Regards.

Reviewer 2 Report
Comments for Authors
- There is no reference in the text to the following literature items: 55, 56, 58, 59, 64, 65, 73, 83, 84
- The authors state that ”In the upper 15 m of this interval (21–180 m), there are still a large number of aragonite, but HMC has almost disappeared” (lines 260-261). But They do not explain why aragonite was preserved and HMC disappeared.
- Moreover in the lines 267-268 the Authors state that ”These characteristics indicate that the unstable primary minerals such as primary aragonite and HMC have all transformed into pure LMC after late meteoric diagenesis”. Thus, the question arises whether aragonite was partially preserved as Authors state in lines 260-261 or, such as HMC, was transformed into LMC. HMC could be preserved if the environmental conditions of carbonate diagenesis are favorable, but aragonite rather not. This problem should be discussed in relation to the presence of REY and diagenetic processes.
- Some of the figures are modified drawings taken from references. This means that the results of the authors' research are only the data included in Table S1.
- In chapter ”3. Expression of REY Parameters” the Authors inform about the method of the La and Ce anomalies calculation (lines 192-196). In chapter ”5.3. Potential of the Xisha Carbonates Preserving Original Seawater REY Signatures” They write about ”positive La anomalies, negative Ce anomalies, and remarkably high Y/Ho ratios” (lines 370-371) but they refer to literature though in the Table S1 the results of element contents measurements are presented. The Authors should comment in more detail the results of their own research.
- The article is very interesting because it presents the problem of the presence of REY in carbonate formations and the possibility of their use in determining the conditions of sedimentation and diagenesis of carbonate formations.However, the Authors focus more on the data from the literature than on the results of their own research, included in Table S1, which has been included as a supplementary material. Some results were not discussed.
- In ”Conclusion” chapter the Authors summarize the data connected with Ce anomaly but there is no information about La anomaly. Is the La anomaly not important?
- The Authors should revise the manuscript, emphasizing the results of their own research and the results in relation to the literature data.

Reviewer 3 Report
Dear Editors and Authors the MS "Rare Earth Element Geochemistry of Late Cenozoic Island Carbonates in the South China Sea Special Issue: Diagenesis and Geochemistry of Carbonates" is a very interesting contribution and i think it merits publication after some moderate revisions.
My main concern or question better is the use of the term Meteoric diagenesis. In my understanding reading the article the authors mean rather weathering process(?). In order to avoid any confusion I would suggest this issue to be clarified in the MS. Diagenesis implies some elevated Temp and Pressure. So if you have meteoric water rich in humic material penetrating downwards its not diagenesis!
Another point is the use of the term "oxidation". Not sure if the authors mean oxic or oxidation. If the second then few parts in the MS are wrong.
I have annotated the main points that the authors shall check in the MS.
Looking forward for an updated version.

Round 2
Reviewer 1 Report
Dear Authors,
I'm glad to know that my comments and suggestions have been useful and have helped you to improve the manuscript. I think it is now ready to be published in Minerals. However, regarding the mineralogical analysis (XRD on powders) I believe that you are right partially. In the Materials and Methods section you should indicate the method exactly. I'm sure that your XRD analysis has been performed on random powders (not oriented!). Also, the addition of an internal standard to the sample powders is strongly recommended for correct quantitative data, because the diffraction peaks (i.e. their form, position, intensity, width, etc ...) can be affected by several things as, for example, the size of crystallites, crystallinity of minerals, the reflection properties of each minerals, and so on.
Best Regards
Reviewer 2 Report
The Authors took into account all suggested corrections. It is good that table 1 was included in the text. However, the conclusions could be developed. However, now the article is much better.
Author Response
Dear reviewer, thank you for your precious comments which have greatly improved the quality of the paper. We have revised the conclusions to make them more concise and clear. In addition, we have made some necessary modifications to the text. Thank you!